# Dynamical Quantum Phase Transitions of the Schwinger Model: Real-Time Dynamics on IBM Quantum

**DOI:** 10.3390/e25040608

**Published:** 2023-04-03

**Authors:** Domenico Pomarico, Leonardo Cosmai, Paolo Facchi, Cosmo Lupo, Saverio Pascazio, Francesco V. Pepe

**Affiliations:** 1Dipartimento di Fisica, Università di Bari, I-70126 Bari, Italy; paolo.facchi@ba.infn.it (P.F.); saverio.pascazio@ba.infn.it (S.P.); francesco.pepe@ba.infn.it (F.V.P.); 2Istituto Nazionale di Fisica Nucleare, Sezione di Bari, I-70126 Bari, Italy; leonardo.cosmai@ba.infn.it (L.C.); cosmo.lupo@poliba.it (C.L.); 3Dipartimento di Fisica, Politecnico di Bari, I-70126 Bari, Italy

**Keywords:** noisy intermediate-scale quantum devices, quantum electrodynamics, dynamical quantum phase transition

## Abstract

Simulating the real-time dynamics of gauge theories represents a paradigmatic use case to test the hardware capabilities of a quantum computer, since it can involve non-trivial input states’ preparation, discretized time evolution, long-distance entanglement, and measurement in a noisy environment. We implemented an algorithm to simulate the real-time dynamics of a few-qubit system that approximates the Schwinger model in the framework of lattice gauge theories, with specific attention to the occurrence of a dynamical quantum phase transition. Limitations in the simulation capabilities on IBM Quantum were imposed by noise affecting the application of single-qubit and two-qubit gates, which combine in the decomposition of Trotter evolution. The experimental results collected in quantum algorithm runs on IBM Quantum were compared with noise models to characterize the performance in the absence of error mitigation.

## 1. Introduction

The availability of noisy intermediate-scale quantum (NISQ) devices in cloud access platforms is a fundamental step towards the quantum computing era. Nonetheless, the limited number of available qubits and the absence of controllable error probabilities prevents these systems from actually outperforming current classical computing capabilities. Multiple hardware setups have been engineered for quantum computing purposes, with different advantages regarding gates’ fidelity and experimental realization. Examples of NISQ devices are represented by circuits with superconducting transmon qubits [1,2,3], ion traps or optical lattices hosting Rydberg atoms [4,5,6,7,8], and qubits encoded in photonic modes of optical setups [9,10], this list being far from exhaustive.

In this framework, high-energy physics represents an interesting testbed for quantum devices. On the one hand, quantum computation can be applied “downstream”, to optimize data analysis and event reconstruction from experiments [11,12,13,14,15]. On the other hand, the “upstream” investigation of gauge theories, especially in their lattice formulation [16,17,18,19,20], can benefit from the possibility of performing quantum simulations of regimes not achievable with perturbative techniques. Long-standing questions related to low-energy processes in quantum chromodynamics (QCD) are still far from the current capabilities of Monte Carlo techniques, due to the sign problem in fermionic amplitudes [21,22,23]. To overcome some of these limitations, research at the interface of quantum information, condensed-matter, and high-energy physics is targeting the adoption of new theoretical and computational tools. The state-of-the-art in the field is represented by tensor network methods, able to reduce the exponential complexity to a polynomial one for states characterized by short-range entanglement [24,25,26,27,28,29,30]. These methods are suitable candidates to obtain a breakthrough in non-perturbative regimes: some preliminary studies about quantum electrodynamics (QED) in one spatial dimension proved the ability of tensor networks in describing a wide phenomenology, such as vacuum phase transition, the string-breaking mechanism, and scattering processes [31,32,33]. On the other hand, highly entangled quantum systems must be studied by means of specifically designed setups, since their complexity cannot be managed with current classical computing capabilities.

A major role in making quantum computation and simulation effective to solve practical problems in NISQ devices is played by error correction and mitigation [34,35,36]. In the context of digital real-time evolution [37,38,39,40,41,42], such procedures should keep the quantum state of the system in the physical subspaces allowed by the gauge constraint. Unfortunately, these techniques are expected to limit the quantum advantage due to the required classical information processing. Nevertheless, current simulations on NISQ devices would benefit from the aforementioned techniques in the description of targeted physical phenomena. The hardest operation for any device is represented by a controlled gate, whose implementation varies in each platform, as with ion traps [4,41,42] or superconducting circuits [37,38,39,40]. These setups elaborate single-qubit states by means of laser pulses, tuned with the targeted transitions. Instead, entangling gates are specifically engineered by exploiting the properties of the hardware, thus leading to crucially different fidelities and times required for the physical implementation. Alternatively, analog simulations, e.g., in optical lattices, can adopt different strategies, such as a periodic drive to obtain energy terms endowed with the same symmetry characterizing lattice QED [43,44]. In this case, errors can induce gauge-invariance-breaking terms, which can lead to an emergent prethermal behavior [45,46,47,48].

This paper aimed at testing the superconducting qubit systems available in the IBM Quantum platform [1] in a simple lattice gauge theory application. We implemented digital evolutions generated by the QED Hamiltonian in 1+1 dimensions, consisting of non-commuting local contributions [4,31,32,33,37,41,49,50,51,52,53,54,55,56] and displaying the occurrence of dynamical quantum phase transitions (DQPTs) [57,58,59] in specific cases of quantum quenches. A discretization of the electromagnetic field is required to implement gauge degrees of freedom in the quantum simulator: accomplishing such a task by replacing the continuous U(1) gauge group with one of the cyclic groups $Z_{n}$ guarantees a unitary implementation of the gauge connections [49]; another possible choice is represented by the family of quantum link models [60]. In order to assign each link of the lattice to a single qubit, we chose a $Z_{2}$ discretization, which ensures a minimal circuit depth [52], thus reducing error sources. The study of real-time dynamics involves three crucial stages: the preparation of initial states, the evolution, and the final measurements. Gate errors affecting a digital simulation accumulate in a more or less coherent manner, which is affected by more variables and more error sources than the stand-alone characterization of gates. Time evolution is partitioned into steps to monitor the measurements statistics’ variation without the inclusion of error correction and mitigation. The ground state preparation required by the chosen quench protocol was specifically designed in order to minimize errors and characterize the first stage’s output statistics. Then, we characterized the effectiveness of time evolution, choosing to analyze the system in proximity of a DQPT, where the dynamics is particularly sensitive to noise [51], thus framing the simulation in most unsafe conditions. A concluding estimation of the amount of error probability reduction required for a partial observation of the targeted DQPT was made by analyzing the statistics of the collected results.

The paper is structured as follows. In Section 2, we introduce the lattice Schwinger model and the Jordan–Wigner transformation that maps it into a qubit system and describe the quench protocol and the DQPTs expected in the model. In Section 3, we describe the experimental scheme composed of ground state preparation and the subsequent Trotter evolution. The collected results are compared with the simulated evolution affected by the error probabilities of noisy gates. In Section 4, we discuss our outcomes in light of the existing literature and present a possible outlook for our research.

## 2. The Lattice Schwinger Model

QED in 1+1 dimensions, also known as the Schwinger model, is a U(1) gauge theory describing the interaction of the electromagnetic field, consisting of only an electric component, and a fermionic particle with mass *m* and charge *g*. The model can be discretized on a one-dimensional lattice with spacing *a*, by associating with each lattice site *x* an anticommuting field $\psi_{x}$, which represents a spinless fermion, while links between each pair of neighboring sites host the gauge degrees of freedom, described by the electric field $E_{x,x+1}$ and the vector potential $A_{x,x+1}$. The latter determines the gauge connection $U_{x,x+1}=e^{iaA_{x,x+1}}$, characterized by the generalized canonical commutation relation $\left[E_{x^{\prime },x^{\prime }+1},U_{x,x+1}\right]=\delta_{x,x^{\prime }}U_{x,x+1}$. The lattice model Hamiltonian for a finite lattice with *N* sites reads [4,31,32,33,49]
(1)$$H=-\frac{i}{2a}\sum_{x=0}^{N-1}\left(\psi_{x}^{†}U_{x,x+1}\psi_{x+1}-H.c.\right)+m\sum_{x=0}^{N-1}{\left(-1\right)}^{x}\psi_{x}^{†}\psi_{x}+\frac{g^{2}a}{2}\sum_{x=0}^{N-1}E_{x,x+1}^{2},$$
where periodic boundary conditions [50,51] require the identification N≡0. The model involves staggered (Kogut–Susskind) fermions [61], described by single-component spinors $\psi_{x}$, with negative-mass components encoded in odd-*x* sites. The physical subspace $H_{G}$ is spanned by states |ϕ〉 satisfying the Gauss law constraint $G_{x}|\phi \rangle =0$ at all sites *x*, where, for a $Z_{n}$ gauge group,
(2)$$G_{x}=\sqrt{\frac{n}{2\pi }}\left(E_{x,x+1}-E_{x-1,x}\right)-\psi_{x}^{†}\psi_{x}-\frac{{\left(-1\right)}^{x}-1}{2}.$$The electric field was simulated in the following through a $Z_{n}$ discretization of U(1) [31,32,33,49,52,53] with n=2. Unlike in the quantum link models [50,51], where the electric field is replaced by a spin operator, the $Z_{n}$ model is based on replacing gauge connections with permutation matrices [49]. In the case of $Z_{2}$, the electric field in each link (x,x+1) can have two eigenstates, which will be labeled as ${\{|\ell =\frac{1}{2}\rangle }_{x},|\ell =-\frac{1}{2}\rangle_{x}\}$, with
(3)$$E_{x,x+1}{|\pm \frac{1}{2}\rangle }_{x}=\pm \frac{\sqrt{\pi }}{2}{|\pm \frac{1}{2}\rangle }_{x},$$
while the gauge connections act as [31,32,33]
(4)$$U_{x,x+1}{|\pm \frac{1}{2}\rangle }_{x}={|\mp \frac{1}{2}\rangle }_{x}.$$An immediate implication of the $Z_{2}$ model is the irrelevance in the Hamiltonian (Equation 1) of the electric field energy, which becomes a constant.

The simplest nontrivial periodic lattice is composed by N=2 sites: the states spanning the physical subspace $H_{G}$ of the $Z_{2}$ model in this simple case are shown in Figure 1. The states in Panels (a) and (d) represent two “Dirac vacua”, with a filled negative-mass and an empty positive-mass site. In these states, the total electric field is constant and nonvanishing. These observations motivated the notation ${|vac\rangle }_{\pm }$ for these two states, where the index is related to the sign of the background electric field. Particle hopping leads to the remaining “mesonic” basis states $|e^{+}e^{-}\rangle_{L}$ and $|e^{+}e^{-}\rangle_{R}$, represented in Panels (b) and (c), respectively, where the index refers to the counterclockwise (L) or clockwise (R) hopping of the fermion from the negative- to the positive-mass site.

The Jordan–Wigner transformation maps the spinor field into a spin system [4,52], which corresponds to our qubit register, as
(5)$$\psi_{x}=\prod_{\ell <x}\left(iZ_{\ell }\right)\frac{X_{x}+iY_{x}}{2},\;\psi_{x}^{†}=\prod_{\ell <x}\left(-iZ_{\ell }\right)\frac{X_{x}-iY_{x}}{2},$$
where *X*, *Y*, and *Z* are Pauli matrices, $\sigma^{\pm }=\frac{X\pm iY}{2}$, and occupied sites correspond to qubit states |↓〉. The Hamiltonian of the resulting spin system is
(6)$$H=H_{J}+H_{m}=\frac{J}{2}\sum_{x=0}^{N-1}\left(\sigma_{x}^{-}U_{x,x+1}\sigma_{x+1}^{+}+H.c.\right)-\frac{m}{2}\sum_{x=0}^{N-1}{\left(-1\right)}^{x}Z_{x}=\sum_{x=0}^{N-1}h_{x},$$
where the free parameter corresponds to a coupling constant $J=\frac{1}{a}$, once energy is scaled in units of mass *m*.

### 2.1. Dynamical Quantum Phase Transitions

We aimed at studying the non-equilibrium dynamics of the described lattice Schwinger model following a quantum quench [50,51,57]. Generally, in this protocol, one considers a family of Hamiltonians H(γ) that depends on a tunable parameter and prepares an initial state coinciding with the ground state $|\psi_{g}\rangle$ of $H_{0}=H\left(\gamma_{0}\right)$. At t=0, the Hamiltonian suddenly switches to $H=H(\gamma_{f})$, determining the evolution $|\psi \left(t\right)\rangle =e^{-itH}|\psi_{g}\rangle$, characterized by the survival (or Loschmidt) amplitude:(7)$$G(t)=\langle \psi_{g}|\psi \left(t\right)\rangle .$$ To identify a possible DQFT, we search for the zeros of the *Loschmidt echo*:(8)$$L\left(t\right)={\left|G\left(t\right)\right|}^{2}=e^{-N\lambda (t)},$$
which depends on the number *N* of degrees of freedom and on the rate function λ(t), which becomes divergent in correspondence with the aforementioned zeros.

The targeted evolution generated by the Hamiltonian (Equation 6) is determined by a single free parameter, as the Loschmidt amplitude phase φ(J,t)=argG(t) is undefined in correspondence with the critical points. Nonetheless, in their neighborhood, φ is expected to be smooth up to a discontinuity line of 2π starting from the criticality. This characterization corresponds to a vortex, with a winding number:(9)$$\nu =\frac{1}{2\pi }\oint_{C}ds\cdot \nabla \varphi ,$$
where C is a loop in the (J,t) plane [50].

The adopted protocol quenches the Kogut–Susskind staggered fermions at t=0 by inverting the mass sign: H(m,J)⟶H(−m,J). The quenched Hamiltonian can be decomposed into parity sectors, as described in Appendix A:(10)$$H\left(-m,J\right)=H^{\left(-\right)}⊕H^{\left(+\right)}.$$ In the even sector, the evolution is generated by
(11)$$H^{\left(+\right)}=\left(\begin{array}{cc} -\frac{J^{2}-m^{2}}{\sqrt{m^{2}+J^{2}}} & \frac{2Jm}{\sqrt{m^{2}+J^{2}}}\\ \frac{2Jm}{\sqrt{m^{2}+J^{2}}} & \frac{J^{2}-m^{2}}{\sqrt{m^{2}+J^{2}}} \end{array}\right),$$
in the subspace spanned by the basis $\{|\psi_{g}\rangle ,|\psi_{\bar{g}}\rangle \}$, made of the even eigenstates of the initial Hamiltonian $H_{0}=H\left(m,J\right)$, which are associated with the lowest and highest eigenvalue $E_{g}=-\sqrt{m^{2}+J^{2}}$ and $E_{\bar{g}}=+\sqrt{m^{2}+J^{2}}$, respectively. The odd parity sector involves the eigenstates $|\psi_{e}\rangle$ and $|\psi_{\bar{e}}\rangle$ of H(m,J), which are independent of *J* and characterized by the eigenvalues ±m; these two states are still eigenstates of the quenched Hamiltonian H(−m,J), which only inverts their eigenvalues.

The Loschmidt amplitude for the initial ground state:(12)$$|\psi_{g}\rangle =a_{g}{(|vac\rangle }_{+}+{|vac\rangle }_{-})+b_{g}{(|e^{+}e^{-}\rangle }_{L}+|e^{+}e^{-}\rangle_{R}),$$
derived in Appendix A, reads
(13)$$G\left(t\right)={\left(2a_{g}^{2}-2b_{g}^{2}\right)}^{2}e^{-iE_{g}t}\left(1+\frac{J^{2}}{m^{2}}e^{-i(E_{\bar{g}}-E_{g})t}\right),$$
with
(14)$$a_{g}=\frac{1}{\sqrt{2\left(1+p_{g}^{2}\right)}},\;b_{g}=\frac{p_{g}}{\sqrt{2\left(1+p_{g}^{2}\right)}},\;\mathrm{with}\;\;p_{g}=\frac{m}{J}-\sqrt{\frac{m^{2}}{J^{2}}+1}.$$ DQPTs are observed for J=m at times
(15)$$t_{j}=\frac{(2j+1)\pi }{2E_{\bar{g}}}=\frac{(2j+1)\pi }{2\sqrt{2}\;m},$$
yielding the Rabi oscillations between $|\psi_{g}\rangle$ and $|\psi_{\bar{g}}\rangle$ expected from Equation (Equation 11), as shown in Figure 2. The behavior of the phase, reported in Figure 2b, features vortices corresponding to Loschmidt echo nodes, while the remaining discontinuities in survival maximum values compensate each other.

### 2.2. Ground State Preparation

The protocol presented in Section 2.1 requires the preparation of the input state $|\psi_{g}\rangle$, namely the ground state of H(m,J). Based on reasons clarified in Section 2.4, the degrees of freedom of the lattice are assigned to the four qubits of the ibmq_manila circuit $|q_{0}q_{1}q_{2}q_{3}\rangle$:$q_{0}$ and $q_{3}$ host the “electric field” states of the $Z_{2}$ links;The staggered spinless fermions are encoded in $q_{1}$ and $q_{2}$.

The four physical states refer to the following computational basis states: ${|vac\rangle }_{-}=|1011\rangle$, $|e^{+}e^{-}\rangle_{L}=|0101\rangle$, $|e^{+}e^{-}\rangle_{R}=|1100\rangle$, and ${|vac\rangle }_{+}=|0010\rangle$, expressed according to the IBM qiskit notation |↑〉=|0〉 and |↓〉=|1〉 for matter sites and $|\frac{1}{2}\rangle =|0\rangle$ and $|-\frac{1}{2}\rangle =|1\rangle$ for links. Since each state can be unambiguously identified by the first two qubits $|q_{0}q_{1}\rangle$, one can associate with the ground state $|\psi_{g}\rangle$ an auxiliary *product* state of two qubits:(16)$$|\psi_{g}^{\prime }\rangle =a_{g}(|10\rangle +\left|00\rangle \right)+b_{g}(|01\rangle +\left|11\rangle \right)=\frac{1}{\sqrt{2}}(|0\rangle +\left|1\rangle \right)⊗\sqrt{2}(a_{g}|0\rangle +b_{g}|1\rangle ),$$
with the amplitudes corresponding, in the DQPT condition J/m=1, to $a_{g}=0.653$ and $b_{g}=-0.271$.

The ground state for the complete four-qubit system is obtained by acting with CNOT two-qubit gates:(17)$$CNOT_{ij}|q_{i}q_{j}\rangle =|q_{i},q_{i}⊕q_{j}\rangle ,$$
which increase the amount of entanglement in the system. For this reason, containing the error probability entailed by these gates is essential to guarantee an effective quantum computation, which cannot be efficiently simulated by classical computers. The circuit chosen for the ground state preparation reads
(18)$$|\psi_{g}\rangle =\;CNOT_{32}\;CNOT_{03}\;CNOT_{13}\;CNOT_{02}\;X_{2}|\psi_{g}^{\prime }\rangle ⊗|00\rangle ,$$
and is pictorially represented in Figure 3.

### 2.3. Noise Models

The simulation of the circuits included an error probability entailed by each gate application, generally described by the bit flip and phase flip error channel $\rho \mapsto D\left[\rho \right]=\sum_{i=0}^{3}K_{i}\rho K_{i}^{†}$, with
(19)$$K_{0}=\sqrt{1-p_{x}-p_{y}-p_{z}}\;𝟙,\;\;K_{1}=\sqrt{p_{x}}\;X,\;\;K_{2}=\sqrt{p_{y}}\;Y,\;\;K_{3}=\sqrt{p_{z}}\;Z.$$ Such quantum channels, explicitly analyzed in Appendix B, are associated with every single-qubit gate employed both in the state preparation and in its time evolution, while for the two-qubit gates, the independent error probabilities $\left(p_{x},p_{y},p_{z}\right)$ can be varied in Equation (Equation 19) to define
(20)$$\rho \mapsto \tilde{D}\left[\rho \right]=\sum_{i,j=0}^{3}{\tilde{K}}_{ij}\rho {\tilde{K}}_{ij}^{†}\;with\;{\tilde{K}}_{ij}=K_{i}⊗K_{j}$$
for a two-qubit density matrix ρ. Each circuit includes also reset and measurement gates, which are affected in the simulations only by bit flips [62,63], implemented by a single noise contribution $K_{1}$, thus corresponding to $p_{z}=p_{y}=0$.

The comparison of the simulations with the outputs of IBM Quantum was evaluated in terms of the trace distance:(21)$$T\left(\rho_{\mathrm{ibmq}},\rho_{\mathrm{sim}}\right)=\frac{1}{2}\left|\right|\rho_{\mathrm{ibmq}}-\rho_{\mathrm{sim}}{\left|\right|}_{1}=\frac{1}{2}\mathrm{Tr}\left[\sqrt{{\left(\rho_{\mathrm{ibmq}}-\rho_{\mathrm{sim}}\right)}^{†}\left(\rho_{\mathrm{ibmq}}-\rho_{\mathrm{sim}}\right)}\right],$$
which quantifies the similarity between the output state of simulated state $\rho_{\mathrm{sim}}$ and the actual output of the IBM hardware $\rho_{\mathrm{ibmq}}$.

### 2.4. Trotter Evolution

The evolution determined by the Hamiltonian (Equation 6), composed of non-commuting local terms $h_{x}$, can be approximated by a Trotter decomposition basis on local unitary operators:(22)$$e^{-iHt}=e^{-i\sum_{x}h_{x}t}={\left(e^{-ih_{N-1}\Delta t}e^{-ih_{N-2}\Delta t}\ldots e^{-ih_{0}\Delta t}\right)}^{\frac{t}{\Delta t}}+O\left(\Delta t\right).$$ The improved approximation that would in principle be provided by the Suzuki–Trotter formula [52] is not well suited in this framework, because it would require a larger number of gates for circuit implementation.

A decomposition of each term in Equation (Equation 22) according to the available set of gates was formulated in [52]. Here, we present its specific application to the $Z_{2}$ gauge group [31,49], where $U_{x,x+1}=U_{x,x+1}^{†}=X_{x,x+1}$. The fermionic hopping contribution can be equivalently expressed as
(23)$$H_{J}=\frac{J}{4}\sum_{x=0}^{N-1}\left[X_{x,x+1}\left(X_{x}X_{x+1}+Y_{x}Y_{x+1}\right)\right]=\sum_{x=0}^{N-1}h_{J,x}.$$ The evolution related to the Trotter time steps generated by the three qubits’ interaction in Equation (Equation 23) was implemented according to the Cartan decomposition [64,65,66]. Concerning the periodic lattice with N=2 sites, we considered for clarity the hopping term $h_{J,0}$, acting on the subsystem $|q_{0}q_{1}q_{2}\rangle$:(24)$$\begin{array}{cc} e^{-ih_{J,0}\Delta t} & =K^{†}AK, \end{array}$$(25)$$\begin{array}{cc} K & =CNOT_{12}\;CNOT_{01}\;H_{1}\;H_{0}\;CNOT_{12}, \end{array}$$(26)$$\begin{array}{cc} A & =𝟙⊗R_{z}\left(J\Delta t/2\right)⊗R_{z}\left(-J\Delta t/2\right), \end{array}$$
where $H_{i}$ is the Hadamard gate acting on $q_{i}$ and $R_{z}\left(\alpha \right)=e^{-iZ\alpha /2}$. The remaining term $h_{J,1}$ acts in an analogous way on the subsystem $|q_{1}q_{2}q_{3}\rangle$, as represented in Figure 4.

The decomposition first rotates the product basis states:



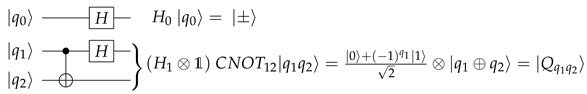



with ${|\pm \rangle =|\left(-1\right)}^{q_{0}}\rangle =\frac{1}{\sqrt{2}}(|0\rangle \pm \left|1\rangle \right)$ the *X* eigenstates. The action of Hadamard gate $H_{0}$ is required to entangle the state of the associated link with matter sites in the following steps: 
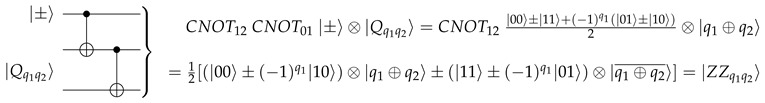

where the bar stands for the logical NOT. A further elaboration of the above $|ZZ_{q_{1}q_{2}}\rangle$ states expression simplifies the application of Cartan decomposition in the evolution with diagonal operators:(27)$$\begin{array}{cc} |ZZ_{q_{1}q_{2}}\left(\Delta t\right)\rangle = & \;A|ZZ_{q_{1}q_{2}}\rangle =𝟙\;⊗\;R_{z}\left(J\Delta t/2\right)⊗R_{z}\left(-J\Delta t/2\right)\left[{|\left(-1\right)}^{q_{1}}\pm \rangle ⊗\frac{1}{\sqrt{2}}\left(|0,q_{1}⊕q_{2}{\rangle +\left(-1\right)}^{q_{1}}|1,\bar{q_{1}⊕q_{2}}\rangle \right)\right]\\ = & {\;|\left(-1\right)}^{q_{1}}\pm \rangle ⊗\frac{1}{\sqrt{2}}\left(e^{-i\frac{J\Delta t}{4}\left(1-{\left(-1\right)}^{q_{1}⊕q_{2}}\right)}|0,q_{1}⊕q_{2}{\rangle +\left(-1\right)}^{q_{1}}e^{i\frac{J\Delta t}{4}\left(1+{\left(-1\right)}^{\bar{q_{1}⊕q_{2}}}\right)}|1,\bar{q_{1}⊕q_{2}}\rangle \right), \end{array}$$
such that states satisfying $q_{1}⊕q_{2}=1$ acquire a time-dependent phase, while there is no evolution outside the physical subspace $H_{G}$ with $q_{1}⊕q_{2}=0$, as predicted by Equation (Equation 6).

Focusing on states with $q_{1}⊕q_{2}=1$, the next circuit steps related to $K^{†}$ yield
(28)$$\begin{array}{cc} |Q_{q_{1}q_{2}}\left(\Delta t\right)\rangle = & \;CNOT_{01}CNOT_{12}|ZZ_{q_{1}q_{2}}\left(\Delta t\right)\rangle \\ = & \;\frac{1}{2}\left(e^{-i\frac{J\Delta t}{2}}|00\rangle \pm {\left(-1\right)}^{q_{1}}e^{-i\frac{J\Delta t}{2}}|11\rangle +{\left(-1\right)}^{q_{1}}e^{i\frac{J\Delta t}{2}}|01\rangle \pm e^{i\frac{J\Delta t}{2}}|10\rangle \right)⊗|1\rangle , \end{array}$$
followed by the last part of the decomposition:(29)$$CNOT_{12}H_{1}H_{0}|Q_{q_{1}q_{2}}\left(\Delta t\right)\rangle =\left\{\begin{array}{c} \cos\frac{J\Delta t}{2}|001\rangle -i\sin\frac{J\Delta t}{2}|110\rangle ,\;\mathrm{if}\;q_{0}=0,q_{1}=0,\\ \cos\frac{J\Delta t}{2}|101\rangle -i\sin\frac{J\Delta t}{2}|010\rangle ,\;\mathrm{if}\;q_{0}=1,q_{1}=0,\\ \cos\frac{J\Delta t}{2}|010\rangle -i\sin\frac{J\Delta t}{2}|101\rangle ,\;\mathrm{if}\;q_{0}=0,q_{1}=1,\\ \cos\frac{J\Delta t}{2}|110\rangle -i\sin\frac{J\Delta t}{2}|001\rangle ,\;\mathrm{if}\;q_{0}=1,q_{1}=1, \end{array}\right.$$
as expected by the action of $e^{-iH_{J,0}\Delta t}$. Depending on states of the remaining link, encoded in $|q_{3}\rangle$, there are states not belonging to $H_{G}$ that show a time evolution: they correspond to the ones shown in Figure 1 with a reversed matter site occupation. The remaining contributions in the Trotter expansion of Equation (Equation 22) are the diagonal mass terms $H_{m}=\sum_{x=0}^{N-1}h_{m,x}$ of Equation (Equation 6), expressed by
(30)$$e^{-ih_{m,x}\Delta t}=R_{z}\left(-{\left(-1\right)}^{x}m\Delta t\right)=e^{i{\left(-1\right)}^{x}mZ_{x}\Delta t/2},$$
as reported in Figure 4.

The topology of ibmq_manila in Figure 5a is well suited for the implementation of the Trotter evolution, since every CNOT involves nearest-neighbor qubits. In a noiseless scenario, the presented Trotter evolution would yield the Rabi oscillations in Figure 2c, corresponding to the analytical solution of the evolution by the quenched Hamiltonian.

## 3. Simulations of Real-Time Dynamics

The experimental results presented in the following were collected in the IBM Quantum platform [1]. The circuit test was based on the simplest periodic lattice required for the implementation of the Schwinger model described in Section 2, composed of N=2 sites for the matter field and an equal number of links endowed with the $Z_{2}$ gauge group. This choice allows for the optimization of the number of gates involved in each Trotter time step, compared to higher-dimensional discretizations of the U(1) gauge group [52]. The simulations include a noise model referring to an error probability affecting each gate, but do not take into account effects related to coherence times, as discussed in Section 4.

The ground state preparation procedure presented in Section 2.2 performs better in terms of fidelity than the Python package qiskit built-in command QuantumCircuit. initialize, as shown in Figure 5. Errors due to the use of CNOT gates between non-neighboring qubits were investigated by implementing the same preparation in two different topologies, shown in Figure 5a,b: in ibm_nairobi, the qubits are encoded as follows (see the qubit labels in Panel (b)): $q_{0}\to “2”$, $q_{1}\to “0”$, $q_{2}\to “3”$, $q_{3}\to “1”$. Actually, Figure 3 shows the presence of three CNOT gates involving $q_{3}$, together with each one of the remaining qubits. For this reason, this is convenient to encoding it into the highest-degree node of Figure 5b, while the ibm_nairobi circuit is limited to the first four qubits. Despite their structural differences, the median values of the fidelities obtained with the ibmq_manila and ibm_nairobi topologies are essentially the same and are about 0.7, showing in both cases a much higher efficiency with respect to the implementation of QuantumCircuit.initialize. However, the collected statistics in Figure 5c, referring to 80 runs for each ground state preparation modality, shows that the interquartile range obtained with ibm_nairobi is smaller than the one provided by ibmq_manila. Moreover, fluctuations towards low values in the former case are much less relevant.

The readout of the output states is based on state_tomography_circuits, which exploits for our four-qubit circuit the Pauli basis, resulting in $3^{4}$ circuits required by the related orthogonal measurements [67]. Simulated noise models include the effects of bit flips in the last measurement part of the circuit [62,63].

The ground state preparation was simulated by defining a noise model in AerSimulator. Three different models were compared with the ibmq_manila output through the trace distance defined in Equation (Equation 21) [68]. Each gate appearing in Figure 3 is affected by error probabilities expressed by the error channels (Equation 19)–(Equation 20). The simulations in Figure 6 use the following models of probability assignment:(a)Single- and two-qubit gates share the same probability parameters $\left(p_{x},p_{y},p_{z}\right)$, generally different along the three axes;(b)Single-qubit gates have the same error probability along each noise direction $p_{1}=p_{x}=p_{y}=p_{z}$; two-qubit gates have an analogous property, but are characterized by an independent probability $p_{2}$;(c)Two parameters $p_{1}$ and $p_{2}$ quantify the error probability along both *X* and *Z* for single- and two-qubit gates, respectively, while errors along *Y* are neglected.

**Figure 6 entropy-25-00608-f006:**
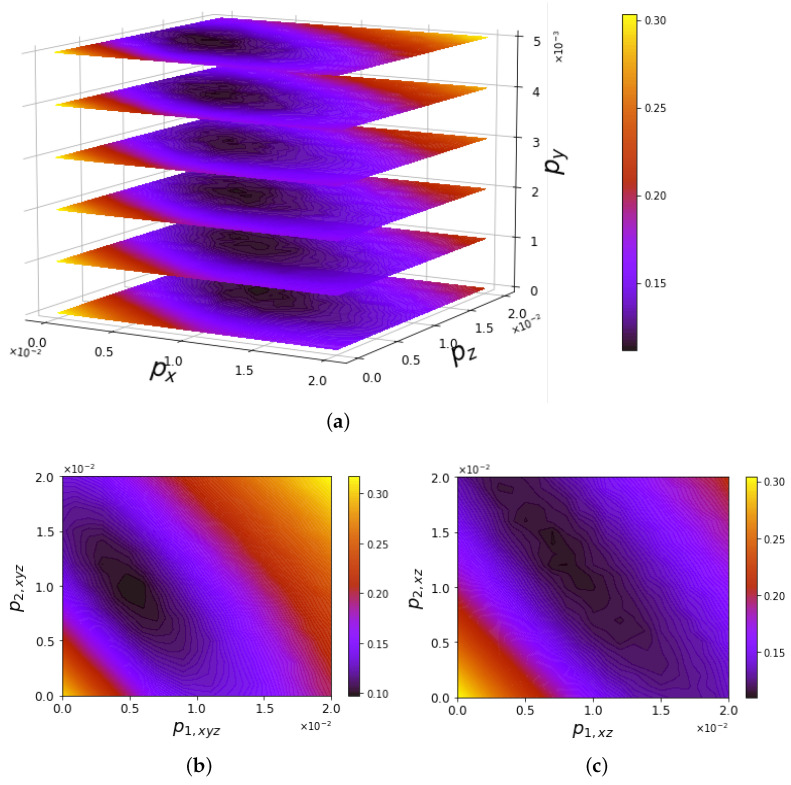
Trace distance $T(\rho_{\mathrm{ibmq}},\rho_{\mathrm{sim}})$ of the simulated noise models affecting circuits’ output with respect to the averaged ground state of ibmq_manila by varying the error probabilities. In Panel (**a**), contour plots in the plane $\left(p_{x},p_{z}\right)$ refer to different values of $p_{y}$. Panels (**b**,**c**) evaluate the variation in terms of single- and double-qubit gate error $\left(p_{1},p_{2}\right)$ with the inclusion or exclusion of *Y* errors, respectively.

The simulations reported in Figure 6a were averaged over 20 realizations of the noise models, while those in Panels (b)–(c) were obtained from 50 realizations. Panel (a) shows a contour plot corresponding to each value of $p_{y}$, obtained by interpolating the trace distance evaluated over a grid consisting of 21×21 points with spacing $1\times 10^{-3}$ in the $\left(p_{x},p_{z}\right)$ plane. The considered output of ibmq_manila is a ground state averaged over 80 realizations. The mean value of the trace distance minimum in the variation of $p_{y}$ is $10.81\times 10^{-2}\pm 3.3\times 10^{-3}$, attained at the averaged coordinates in the plane $\left(p_{x},p_{z}\right)=\left(\left(10.7\pm 1.1\right)\times 10^{-3},\left(6.3\pm 0.9\right)\times 10^{-3}\right)$. These results are stable with respect to variations in $p_{y}$. Indeed, a *Y* error is provided by a sequence of simultaneous bit flips and phase flips, whose probability is much smaller than that of each single error. Panels (b)–(c) show the trace distance of the two noise models with independent parameters for single- and two-qubit gates with respect to the ground state $\rho_{\mathrm{ibmq}}$ experimentally prepared 80 times, with no error along *Y* in the case of Panel (c). The minimum values of the trace distance in the two cases can be considered equal within the statistical fluctuations. Moreover, the minima are found in correspondence with $\left(p_{1,xyz},p_{2,xyz}\right)≃\left(5\times 10^{-3},1\times 10^{-2}\right)$ in Panel (b) and $\left(p_{1,xz},p_{2,xz}\right)≃\left(7.5\times 10^{-3},1.5\times 10^{-2}\right)$ in Panel (c), highlighting a scale factor of 3/2 due to the absence of the error along *Y* in the latter case. The minimum values of the trace distance, which equal $9.77\times 10^{-2}$ in Panel (b) and $11.16\times 10^{-2}$ in Panel (c), express the ability to distinguish the output $\rho_{\mathrm{sim}}$ from the experimental one $\rho_{\mathrm{ibmq}}$ approximately once out of ten times.

The ground state in the input evolves according to the quench protocol presented in Section 2.1. Non-commuting local terms in the Hamiltonian, determined by the Jordan–Wigner transformation (Equation 6), are circumvented by means of the Trotter evolution described in Section 2.4, which was implemented using the ibmq_manila topology. The Loschmidt echo and the overlaps with the remaining physical states are shown in Figure 7, where the different curves were obtained by varying the time step length Δt and by averaging 10 experimental realizations of evolution for each case.

The high error probability translates into a fast convergence towards the maximally mixed state $\rho_{\infty }={\left(\dim\;H\right)}^{-1}𝟙$. This trend shows a striking deviation for the probability ${\left|\langle \psi_{e}|\psi \left(t\right)\rangle \right|}^{2}$ during the first two time steps, probably driven by a coherent error accumulation. To focus on this behavior, as well as to limit the computational resources required in simulations, the trace distance is averaged over the first three time steps $t_{i}=\left(i-1\right)\Delta t$, using
(31)$$\bar{T(\rho_{\mathrm{ibmq}},\rho_{\mathrm{sim}})}=\frac{1}{3\Delta t}\sum_{i=1}^{3}T\left(\rho_{\mathrm{ibmq}}\left(t_{i}\right),\rho_{\mathrm{sim}}\left(t_{i}\right)\right)\Delta t,$$
expressing the mean probability of distinguishing the evolution outputs of ibmq_manila from the simulated ones.

The optimal error probabilities for the ground state preparation are identified in Figure 6c. Simulations of the evolution including noise models fix these parameters assigned to the gates in Figure 3. The Trotter evolution given in Figure 4 includes gates affected by error probabilities, which we implemented according to the following two models:Single- and two-qubit gates are characterized by the same arbitrary parameters $\left(p_{x},p_{z}\right)$;Equal error probabilities in the *X* and *Z* directions, but they take generally different values for single-qubit gates ($p_{1}=p_{x}=p_{z}$) and two-qubit gates ($p_{2}=p_{x}=p_{z}$).

The time step that determines the best resolution trade-off in view of investigating the DQPT point is Δt=0.1. In the averaged trace distance (Equation 31), the argument $\rho_{\mathrm{ibmq}}$ is averaged over ten experimental realizations of the density matrix evolution, as represented in Figure 7, while $\rho_{\mathrm{sim}}$ is averaged over 20 noise realizations for each time step of the evolution. The contour plots in Figure 8 interpolate the evaluation over a grid composed by 31×31 points with spacing $10^{-3}$. They describe how distinguishable both aforementioned noise models are with respect to the evolution in ibmq_manila. The minimum value for both models in Panels (a)–(b) is obtained by a further evaluation along the elongated direction, and it is approximately equal to $23.5\times 10^{-2}$, corresponding to $\left(p_{x},p_{z}\right)≃\left(1.1\times 10^{-2},1.5\times 10^{-2}\right)$ in the case of Panel (a) and $\left(p_{1},p_{2}\right)≃\left(1\times 10^{-2},1.6\times 10^{-2}\right)$ in the case of Panel (b). These values are used in Figure 9 to compare the evolution of physical states. In Panels (a)–(d), the overall behavior characterized by a convergence towards the maximally mixed state is captured by the noise models, but non-negligible deviations corresponding to the probability ${\left|\langle \psi_{e}|\psi \left(t\right)\rangle \right|}^{2}$ in Panel (c) are still not captured by the model, probably because they are driven by coherently accumulated errors, not included for stand-alone gates in current noise models’ implementation. A comparable high number of single- and two-qubit gates makes the two noise models overlap, as represented in Figure 8, so we can focus on the simpler one described by the probabilities $\left(p_{x},p_{z}\right)$. Such a probability array is denoted as p in Figure 9e–h, where the comparison of the reduced noise regimes aimed at estimating a threshold such that a revival is observed after the first DQPT. This translates into the research of a non-monotonic behavior of the Loschmidt echo, which requires an overall error probability per gate ten-times lower than in current implementations.

## 4. Discussion

Error correction and mitigation techniques are crucial for NISQ devices in order to efficiently simulate the targeted dynamics [34,35,36]. An example was given in [37], concerning the real-time dynamics on IBM Quantum of a periodic lattice model for a 1+1 QED model with N=4 matter sites. There, the discretization of the gauge degrees of freedom was based on a different (non-unitary) truncation. Moreover, the targeted quantities were the vacuum energy and pair production, without a focus on the observation of DQPTs. The exploitation of parity sectors and allowed momenta entails a large reduction of the required qubit number, yielding a scheme able to constrain the evolution in the physical subspace. The reduced Hamiltonian for the matter degrees of freedom in the targeted sector generates a Trotter dynamics implemented in circuits through the Cartan decomposition. Aiming at the zero noise extrapolation, a procedure adopting repeated application of noisy CNOT gates, was implemented. The circuit depth for each time step increases with respect to the decomposition described in this work, thus allowing us to estimate our targeted first DQPT (with time step Δt=0.1) corresponding to the $T_{2}$ coherence time, because of the 10-time-step limit mentioned in [37]. Indeed, the total number of circuit moments in our Trotter evolution with 10 steps is equal to 200, as shown in Figure 4. We have to include the ground state preparation depth in Figure 3, because of the SWAP gates related to three CNOT between non-nearest neighbor qubits. The maximal gate temporal extent allowed by the coherence time is slightly lower than the effective one, thus signaling an overestimation of our error probabilities.

The comparison of our results with an ion trap simulation of the lattice Schwinger model with N=4 matter sites in [4] has to take into account the much lower number of Mølmer–Sørensen gates, determined by the higher value of the time step. This is related to the different purpose of the aforementioned work, which aimed at characterizing the pair production mechanism.

The evolution in the proximity of a DQPT is considerably more affected by noise [50,51], as simulated for a transverse field Ising model [40] with error rates comparable to those obtained in our study. Nonetheless, the Hamiltonian terms of an Ising model concern at most spin pairs, thus reducing the circuit complexity for Trotter product formulas. In the case of commuting Hamiltonian terms, at the basis of plaquette dynamics without matter degrees of freedom [38], the Trotter product is not required, thus yielding a further reduction of the circuit depth. Concerning the estimated error probabilities in our analysis, their magnitudes were confirmed in the study of scalar Yukawa coupling [69].

The current experimental realization of ion traps and Rydberg atoms in optical lattices shows a higher value for the average gate fidelity of entangling gates [5,70,71]. The introduction of thermal effects, as well as an increased number of parameters for gates’ errors must be considered in order to improve the proposed noise models. Such a detailed description is motivated by the determined “optimal” error probabilities for ground state preparation, which do not coincide with those characterizing time evolution. Moreover, the coherent error accumulation would require more sophisticated noise models, such as correlated dissipation in subsequent quantum channels. The inclusion of error correction and mitigation [37,38,39,40] will be investigated in future research to keep the dynamics in the physical subspace and to balance the noise affecting DQPTs’ observation.

## 5. Conclusions

We studied the possibility of simulating the real-time dynamics of a model of QED in 1+1 dimensions, on an elementary lattice composed of two fermionic sites, implemented on IBM Quantum [1]. More specifically, we analyzed the dynamics after a mass quench, close to a dynamical quantum phase transition. The considered quench protocol requires ground state preparation, based on an optimized circuit able to outperform built-in functions, as measured by fidelity with the ideal state. Limitations in observing DQPTs were described in terms of the error probabilities associated with each gate. Different noise models were simulated and compared to capture the main features of the measured evolution, thus determining a marginal contribution of noise along the *Y* direction. These minimal models revealed the partial observation of the targeted DQPTs’ phenomena in circuit implementations with a reduced error probability. The estimated error rate also indicated a promising implementation on ion traps, such as those available in the IonQ platform [72].

## Figures and Tables

**Figure 1 entropy-25-00608-f001:**
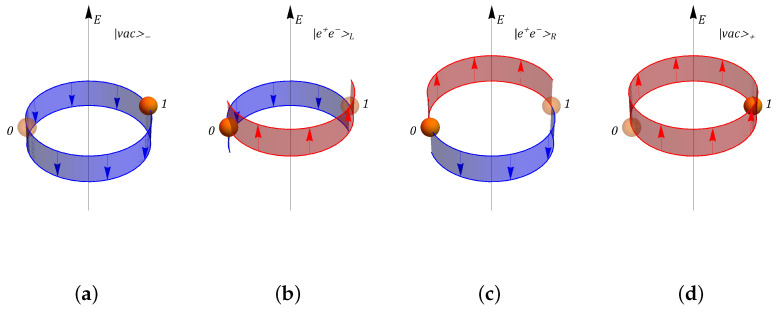
Representation of the physical subspace basis states in a $Z_{n}$ gauge model, implemented on a two-site lattice. In all panels, full (transparent) spheres represent occupied (empty) matter sites, while red (blue) edges correspond to a positive (negative) electric field on a link. Panels (**a**,**d**) represent “Dirac vacua”, characterized by an occupied negative-mass fermion site, an empty positive-mass fermion site, and a constant background electric field. Panels (**b**,**c**) represent “meson” states, with an occupied positive-mass fermion site, an empty negative-mass fermion site (corresponding to an antiparticle), and a staggered electric field.

**Figure 2 entropy-25-00608-f002:**
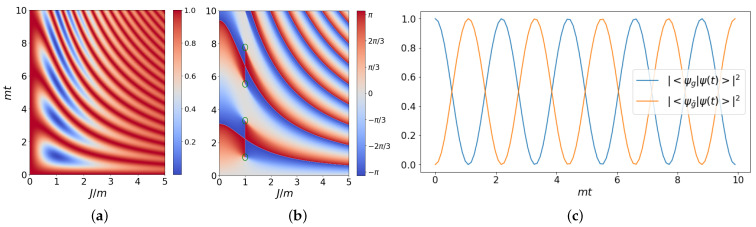
Panel (**a**) shows the Loschmidt echo $L\left(t\right)={\left|\langle \psi_{g}|\psi \left(t\right)\rangle \right|}^{2}$ with the variation of the free parameter *J*, while the phase of the Loschmidt amplitude is represented in Panel (**b**); here, the green paths around the DQPT points are characterized by a nonvanishing winding number. The Trotter evolution discussed in Section 2.4 of Rabi states with step Δt=0.1, corresponding to J/m=1 and without noise, is shown in Panel (**c**).

**Figure 3 entropy-25-00608-f003:**
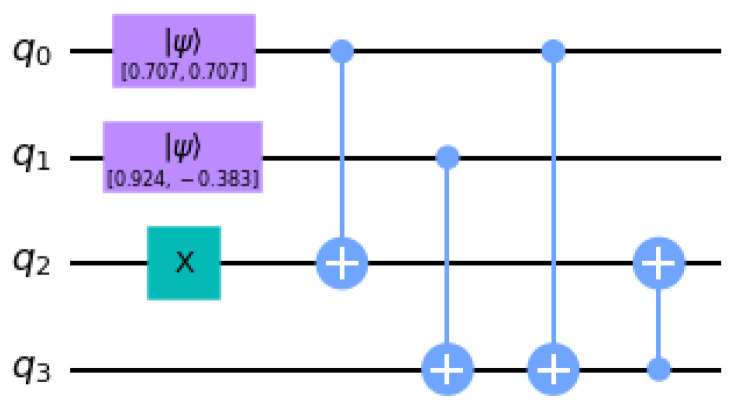
Circuit for the ground state preparation of H(m,J) corresponding to the dynamical quantum phase transition value J/m=1. Matter sites correspond to $q_{1}$ and $q_{2}$, while $Z_{2}$ links are encoded in $q_{0}$ and $q_{3}$.

**Figure 4 entropy-25-00608-f004:**
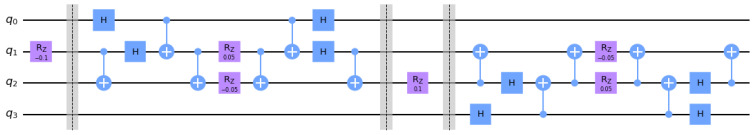
Trotter step as given in [52] for the $Z_{2}$ gauge group discretization of lattice QED. Gauge degrees of freedom are encoded in qubits $q_{0}$ and $q_{3}$, while fermionic matter is described by qubits $q_{1}$ and $q_{2}$. The parameters used in $R_{z}$ gates correspond to the choice J=m=1 and Δt=0.1.

**Figure 5 entropy-25-00608-f005:**
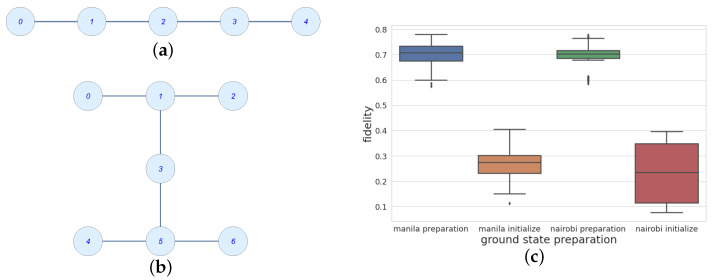
Topologies of the circuits ibmq_manila, in Panel (**a**), and ibm_nairobi, in Panel (**b**). The distributions of the ground state preparation fidelity obtained by applying the scheme proposed in Section 2.2 are compared with the output of the built-in command QuantumCircuit.initialize in the boxplot of Panel (**c**).

**Figure 7 entropy-25-00608-f007:**
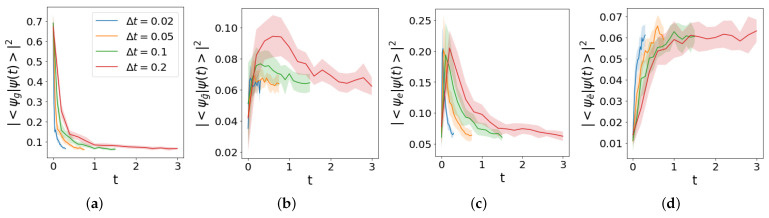
Trotter evolution of the initial state $|\psi_{g}\rangle$ for different time steps Δt, run in ibmq_manila. The average over 10 realizations corresponds to the solid line, while the shaded region represents the standard deviation. Loschmidt echo ${\left|\langle \psi_{g}|\psi \left(t\right)\rangle \right|}^{2}$ is shown in Panel (**a**), while probabilities of finding the evolved state in remaining physical states $|\psi_{\bar{g}}\rangle$, $|\psi_{e}\rangle$, $|\psi_{\bar{e}}\rangle$ are reported in Panel (**b**),(**c**),(**d**) respectively. Time is expressed in units of $m^{-1}$ in all plots. The corresponding noiseless evolution is reported in Figure 2c.

**Figure 8 entropy-25-00608-f008:**
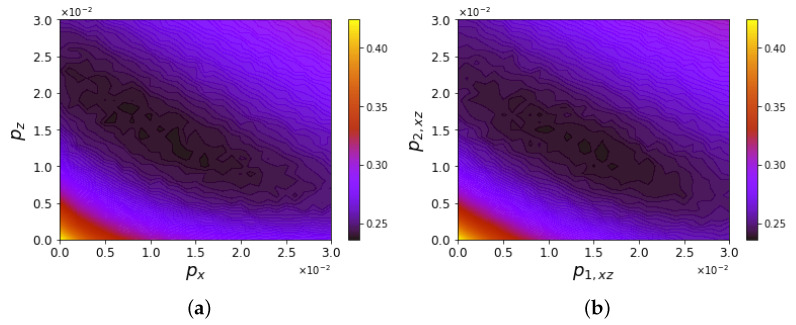
Averaged trace distance $\bar{T(\rho_{\mathrm{ibmq}},\rho_{\mathrm{sim}})}$ of the evolution yielded in the first three time steps by ibmq_manila and the simulated noise models. In Panel (**a**), probability parameters $\left(p_{x},p_{z}\right)$ are related to the *X* and *Z* errors, while, in Panel (**b**), $\left(p_{1,xz},p_{2,xz}\right)$ refer to the error probabilities in the single- and two-qubit gates, respectively, in the case in which *Y* errors are neglected.

**Figure 9 entropy-25-00608-f009:**
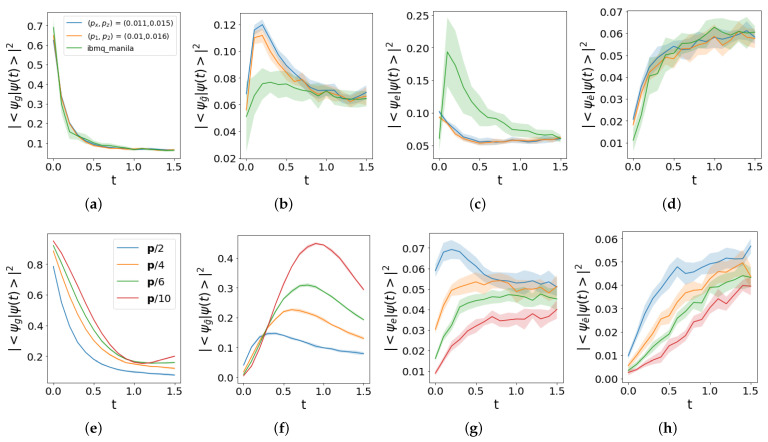
Noise model implemented in the qiskit simulation with Trotter step Δt=0.1. In Panels (**a**–**d**), we use the optimal error probabilities determined in Figure 6, Figure 7 and Figure 8. In Panels (**e**–**h**), such a probability array is denoted by p to compare different noise regimes in order to observe a revival following the first dynamical quantum phase transition. In all plots, time is expressed in units of m${}^{-1}$.

## Data Availability

The data presented in this study are available upon request from the corresponding author.

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
