# Peer review of "Dynamical Quantum Phase Transitions of the Schwinger Model: Real-Time Dynamics on IBM Quantum"

_entropy, 2023, doi:10.3390/e25040608_

Round 1
Reviewer 1 Report
This paper reports the application of quantum computation on the simple gauge model: (1+1)D Schwinger model on a lattice site. The electric field degree of freedom is described by descritized values n=2. Based on the Kogut–Susskind Hamiltonian, the excitation of particles are registered as spin down at odd site as the excitation of an electron, while spin up state at even lattice site as the excitation of a positron. The electric field may take the integer numbers up to a cut-off value. Similar works have been done and reported in Ref[4] and Ref[37].
The authors mention in line 49 "This paper is aimed at testing the superconducting qubit systems available in the IBM Quantum platform in a simple lattice gauge theory application." As such I think the introduction is not well-written for such a motivation. It seems to me that the authors used IBMQ to test how good is the NISQ with a simplest gauge theory calculation. Therefore, paragraph 2 is not necessary since this work does not emphasize the computation capacity of the quantum computer. Instead, the authors need to review how NISQ limits the quantum computation and simulations in previous trials. In view of this, I suggest to move some part of in Discussion section to the introduction and cite more works in this regard.
In particular, gauge theory calculations that applies ion-trap QC (Ref[4]) and IBMQ (Ref[37]) have been reported. A clear comparison in the calculation error using different platforms would be important and interesting.
Moreover, there is no clear introduction of the method in simulating the NISQ calculations. Please provide some background knowledge on the method and package you used. The assumptions on the simulation are mandatory for judging how good the simulator is. Regarding this, I only found one sentence: "Simulated noise models include the effects of bit flips in the last measurement part of the circuit [56,57]." I wonder whether this kind of noise model sufficient for estimating your quenching algorithm results? Is there any other noise model to use?
Because of the major problems in the presentation, I would judge that the manuscript is not suitable for publication without major revision.
Here I also raise some other questions for the authors to clarify:
-Since the novelty of this work is not to implement a gauge theory to a practical QC and the implementation has been reported in previous works, I feel that the authors can cite the references and shorten or move Sec. 2.1 and 2.2 to the appendix. It will be more important to see how the preparation and quenching evolution of the gauge model(e.g. Fig. 3 and Fig. 4) are conducted in IBMQ.
-There is no show of the error-free results as a comparison in Figs. 7 and 9.
-Are the "optimal" error probabilities for the ground state preparation the same of the optimal error probabilities in the following up calculations? If yes, why the state preparation can be used to quantify the optimal error probabilities, instead of some other algorithms?
-In Fig.9(a), the ibmq data and the simulated results are quite different. Please give some comments on the possible reasons.
-It is better to note each panel in Fig. 7 (as well as Fig. 9) individually from (a), (b) .... (d).
-In line 141, "The circuit test is based on the simplest periodic lattice required for the implementation of the Schwinger model....This choice allows for the optimization of the number of gates involved in each Trotter time step, compared to higher-dimensional discretizations of the U(1) gauge group [50]."
It helps the readers a better understanding if mentioning in the introduction that n=2 is chosen for your work, and to provide some information what do you expect to see with the QC.
-The notations on vacuum states are ambiguous. Sometimes it is noted as |vac>(+-) and sometimes |vac>(0,1) (in Fig. 1 and (A.1))
-The notation in line 78(also in line 112) for the E-field states is easily confused with the registration of particle states as noted in previous literatures. Since it only appears twice, it is better to note clearly and unambiguously. For example (|l=-1/2> and |l=1/2>) as in Ref[4].
-The ground state preparation is simulated by defining a noise model in "AerSimulator" with probability parameters (px, py, pz). Please provide some background knowledge for this noise model and parameters.
-In Fig. 2, the authors present the expected time evolutions of the survival amplitude, and note as the important feature of the problem they are study. From the practical QC result, it is very difficult to see there is "Loschmidt echo". Could the simulator produce such an echo when the error probability is small? If yes, what is the critical value for displaying the echo?
Author Response
A point-by-point response is available in the file "reply.pdf".

Reviewer 2 Report
Comments are included in the attached file

Author Response

(The authors gave the same response as above.)

Round 2
Reviewer 1 Report
The revised version is satisfactory and I recommend its acceptance.
One minor thing to check again:
Should " |e+e−⟩L 1⟨vac| + |e+e−⟩R 1⟨vac| " in Eq.(A1) be
"|e+e−⟩L +⟨vac| + |e+e−⟩R +⟨vac|"?